# Dieters in the Covid-19 Pandemic: Risk for Eating Disorders and Their Association with Food Cravings and Intuitive Eating Traits

**Jônatas Oliveira** [1,*] , **Isis de Carvalho Stelmo** [2] , **Mariana Bueno Netto Santaella** [1] and **Táki Athanássios Cordás** [1]

1   Eating Disorders Program (AMBULIM/PROTAD), Institute of Psychiatry, University of Sao Paulo,
    Sao Paulo 01060-970, Brazil; maribuenonetto@gmail.com (M.B.N.S.); cordas@usp.br (T.A.C.)
2   Department of Psychology, University of Roehampton, London SW15 5PJ, UK; nutriisisstelmo@gmail.com
*   Correspondence: oliveira.jonatas@usp.br

**Abstract:** Objectives: To characterize a group of university students in social isolation regarding their eating behaviors and to investigate whether dieters have a tendency to engage in binge-eating. Study Design: Cross-sectional study carried out during Brazil's first months of social isolation. Methods: University students filled out the Eating Attitudes Test—EAT, the Binge-eating Scale—BES, and the Hay questionnaire. In addition, the current research also evaluated food cravings (FC) and intuitive eating. Student's t-test was used to compare the effect sizes (Cohen's d) between groups. Results: More than 90% of participants were isolated from academic activities, and 68% reported significant dietary change. Those who dieted ($n = 57$) were less confident in signs of hunger and satiety ($d = -0.9$, $p < 0.001$) and had higher binge-eating and FC levels than those who did not diet. Conclusions: Data reinforces the high prevalence of risk for eating disorders in university students, the impact of dieting on FC, and intuitive eating during social isolation.

**Keywords:** dieting; food craving; binge-eating; eating disorders; COVID-19





## 1. Introduction

The COVID-19 pandemic hit Brazil in March 2020, affecting the mental health and eating quality of university students, who had their routines modified in several manners. Social isolation significantly altered people's psychosocial aspects, which was reflected in modified eating and exercise patterns. People with Eating Disorders (EDs) were negatively impacted by this secluded period. Simone and colleagues reported a high number of binge-eating episodes and emotional eating during social isolation in individuals with depression symptoms [1]. Haghshomar et al. developed a meta-analysis in which they highlighted binge-eating increases in almost 50% of participants, and 36% of subjects were found to have adopted unhealthy compensatory strategies [2]. According to the authors, eighty percent of the sample showed greater concerns over weight gain and eating. Rodgers and colleagues also highlighted the effects of exposure to "anxiety-provoking media"—such as diet advertisements—on body image, which culminates in emotional distress and reactive emotional eating [3]. As a result, there has been a decrease in protective factors whilst there has been an increase in the access to adaptive forms of dealing with difficult emotions, stemming from a lack of social support during isolation. This could be associated with a higher risk of developing EDs.

Few studies have focused on the effects of isolation on eating behavior. According to recent data, Brazilians with a previous diagnosis of depression had a higher risk for unhealthy eating behaviors [4]. In addition, Brazilian data between June and September 2020 indicated that "snacking" and desire for food is associated with increased consumption of sweet beverages and highly processed foods, according to NOVA classifications [5–7]. Such behaviors, often classified as unhealthy eating habits, work as mechanisms for dealing with dysregulated emotional states [8], and the desire to eat acts like a bridge between

emotional dysregulation and searching for food to cope with it. Individuals who use food as an emotional regulation (ER) mechanism—for example, to relieve boredom—have shown higher disconnection to body signs [9]. Other study showed heightened exercise patterns in individuals with ED, attributed to possible concerns with body changes and weight gain during social isolation [10]. Changes in diet and exercise patterns could work as a form of emotional regulation [11]. Thus, there are behavioral profiles prone to dieting as an ER mechanism for enduring the impact of social isolation [4,5].

In addition, little is known about the health outcomes and social consequences resulting from isolation—for instance, a lack of social support [12]. For individuals at risk for eating disorders (EDs), it is essential to verify the propensity to engage in behaviors to modify body weight (i.e., diets and inappropriate compensatory practices) [3]. Concomitantly, an overall increase in risk for EDs and decrease in protective factors were observed.

EDs represent a category of severe psychiatric conditions that affect eating behavior and compromise psychosocial functioning. Dieting during social isolation can increase the risk of ED or worsen the disease [3]. However, few studies have evaluated the impact of the pandemic on the context of behaviors concerning food choices and pre-swallowing behavioral aspects such as food cravings [5]. This study aimed to characterize the prevalence of dieting practices and to verify if dieters showed a greater risk for ED compared to non-dieters during the pandemic.

## 2. Study Design

### Characterization of Study Participants

This study aimed to analyze eating behaviors and pre-deglutition aspects, focusing on risks for the development of EDs and the severity of this phenomena associated with a disconnected food intake—namely, altered food cravings, binge-eating, and uninhibited eating. It does not include outcomes regarding food consumption evaluations [13].

This cross-sectional study was conducted remotely between 5 May and 10 August 2020, when activities at universities were suspended and people were in self-isolation. The participants were male and female Brazilian students aged between 18 and 60 years old enrolled in undergraduate and postgraduate courses. The study was approved by the University's Institutional Review Board (n. 2.695.532) and all participants signed a consent form via a link before accessing the questionnaire. Underage participants, and those who had answered the four catch questions in the form incorrectly—namely, "how much is 10 + 2", "this question is to get your attention", "are you filling in attentively?" and "how much is 1 + 5"—were removed from the study. To ensure no data loss, the online questionnaire system checked when questions were not answered.

## 3. Instruments

General data and participant characteristics were collected using a structured questionnaire, amongst other questions aiming to comprehend the relationship between food context and the pandemic. The survey contained closed questions with "yes" or "no" answers and a frequency questionnaire proposed [14] for the evaluation of disordered eating symptoms in intermittent fasting practices. Aiming to understand the food scenario and its relationship with the pandemic, questions were created to investigate diet practices, weight/body satisfaction, and weight loss plans:

1. Currently, during quarantine, has your diet changed significantly? ('yes', 'no')
2. Currently, during quarantine, have you been on a diet? ('yes', 'no')
3. Do you practice intermittent fasting to lose weight or to not gain weight? (never, less than once a week, once a week, twice a week or more times a week)
4. Do you have plans to lose weight in 2020? ('yes', 'no')
5. Did you weigh yourself before the quarantine started? ('yes', 'no')
6. Did you weigh yourself during the quarantine? ('yes', 'no')
7. How satisfied, from 0 to 10, are you today with your weight/body? (0–10)
8. Are you currently attending classes in person on campus? ('yes', 'no')

### 3.1. Questionnaires for the Assessment of Disordered Eating Behaviors

The instruments validated in Brazil and the calculated reliability coefficients (Cronbach's $\alpha$) are described as the following:

The Eating Attitudes Test (EAT-26) was used to assess the risk of ED, with a score > 20 being considered a risk ($\alpha = 0.83$) [15,16]. The presence of binge-eating in scores >17 was assessed with the Binge-eating Scale (BES, $\alpha = 0.91$) [17]. Higher scores on the BES indicate a higher severity of binge-eating. The frequency of inappropriate compensatory behaviors (laxatives, diuretics, and self-induced vomiting) was also evaluated, considering the three-month period prior to the questionnaire [14]. According to the DSM-5, at least one type of compensatory behavior is necessary for a diagnosis of ED.

### 3.2. Criteria for Eating Disorder Symptoms

According to the BES, binge-eating (>17 points) and inappropriate compensatory behaviors indicate Bulimia Nervosa behaviors. The combination of questionnaires to verify the association or lack thereof between binge-eating–purging proposed in this investigation was tested in a previous publication [18]. The presence of binge-eating not associated with inappropriate compensatory behaviors indicates binge-eating behaviors.

### 3.3. Intuitive Eating Scale and Food Cravings Questionnaire

The Intuitive Eating Scale (IES-II) was applied to assess thoughts and behaviors associated with "unconditional permission to eat", "eating for physical rather than emotional reasons", "reliance on hunger and satiety cues", and "body-food choices congruence". Higher scores indicate intuitive eating behaviors, meaning they are aligned with the state and physiology of the body [19]. To explore levels of food cravings (FC), a short version of the FC questionnaire—the trait version (recurrent behaviors and cognitions, $\alpha = 0.92$)—and FC in their momentary aspect (state, $\alpha = 0.91$) were also evaluated. High scores on the scales indicate higher FC [20].

### 3.4. Criteria for Selecting Groups

Considering that having plans to lose weight recruits cognitive restraint, planning strategies, and diet mentality mechanisms, the fourth item in the questionnaire aimed to differentiate those that had plans to lose weight from those who had no plans to lose weight. This question also considered changes in planning patterns as a result of the quarantine during the pandemic. The groups were then selected based on the question, "Now, during quarantine, have you been on a diet?" ((i) Was on a diet and had plans to lose weight; (ii) Was not on a diet and had no plans to lose weight)), disregarding individuals who were not on a diet and had plans to lose weight after the comparison analysis. The removal of individuals with intentions to lose weight from the analysis occurred because having plans to lose weight does not necessarily guarantee actions and behaviors will change—just as cognitive restraint (planning and knowing how to eat in order to lose weight) does not guarantee lower consumption in terms of the amount of food or caloric intake [21].

### 3.5. Data Analysis

In order to verify distribution distortions, we analyzed the normality of the variables according to asymmetry and kurtosis parameters, from 2.0 to 7.0, respectively. We then proceeded with the descriptive analysis and the parametric $t$-tests for the comparison of means. Finally, the effect sizes were calculated by calculating Cohen's d (JASP software, Version 0.12.2, Amsterdam, Netherlands).

## 4. Results

Initially, 458 participants were included in the study, and after the application of the exclusion criteria, 17 participants were removed during the alert questions, 27 were excluded for not belonging to the university, 8 for being underage, and 1 questionnaire was duplicated.

Thus, the study included 405 individuals with a mean age of 23 years (18 to 43; SD = 4.24) and a BMI of 22.3 (SD = 4.0)—353 females (87%) and 52 males (12%)—who reported a high prevalence of isolation (96%) and significant dietary change (68%; Appendix A). Half of the participants reported some binge-eating behaviors, with differing frequencies. In addition, the study found a higher risk of EDs according to EAT in 82% of dieters.

Considering that planning to lose weight results in cognitive restraints and concerns about food and body image [22], in the present study, those who 'were not on a diet but had planned to lose weight' (*n* = 204; 50%) were removed from the comparison analyses (Table 1). The sample was divided into two distinct groups: **(i) those who were on a diet and had plans to lose weight** (*n* = 57; 14% of the sample), **(ii) those who were not on a diet and had no plans to lose weight** (*n* = 144; 36%).

**Table 1.** Comparisons of eating behavior parameters as a function of diet practices.

| | Were on a Diet and Had Plans to Lose Weight (*n* = 57) | Were not on a Diet and Had no Plans to Lose Weight (*n* = 144) | Student's *t* Test | |
|---|---|---|---|---|
| | Mean (SD)/Min-Max | Mean (SD)/Min-Max | *t* | Cohen's d |
| BES | 15.6 (10.1)/1–40 | 7.6 (6.4)/0–38 | 6.717 | 1.050 * |
| EAT | 28.4 (10.4)/7–53 | 11.9 (6.8)/6.8–43 | 13.093 | 2.047 * |
| Food Craving-Trait | 26.2 (9.6)/9–46 | 20.6 (7.7)/8–45 | 4.281 | 0.669 * |
| Food Craving-State | 11.7 (5.0)/5–21 | 10.13 (5.0)/5–25 | 2.025 | 0.317 * |
| Body satisfaction (0–10) | 3.9 (3.0) | 6.7 (2.2) | −7.324 | −1.145 |
| Intuitive Eating | 69.6 (6.4)/59–88 | 72.3 (6.7)/57–93 | −2.587 | −0.404 * |
| Permission to eat | 17.7 (2.1)/12–23 | 17.7 (2.5)/12–26 | 0.026 | 0.004 |
| Eating for physical reasons | 24.4 (2.7)/19–31 | 22.6 (2.7)/13–29 | 4.131 | 0.646 * |
| Reliance on hunger/satiety | 16.9 (4.60/8–30 | 21.53 (4.5)/8–30 | −6.393 | −0.999 * |
| Body-food choice congruence | 10.4 (1.9)/7–15 | 10.3 (2.2)/4–15 | 0.204 | 0.032 |
| Bulimia symptoms | *n* = 3 | *n* = 1 | - | |
| Binge-eating symptoms | *n* = 8 (14%) | *n* = 12 (8%) | - | |
| EAT > 20 | *n* = 47 (82%) | *n* = 16 (11%) | - | |

Legend: * = *p* < 0.05 for comparisons between groups; SD: standard deviation; degrees of freedom for all the comparisons: 200; **Note**: Presence of binge-eating according to cut-off point (17) in Binge-eating Scale.

Additionally, behaviors associated with a concern for the body were noticed, such as weighing during quarantine in 44.9% and fasting to lose weight in 43% (*n* = 174) of participants—even though 59% of participants had a normal weight according to their BMI. Dieting was associated with higher FC levels (d = 0.66 for Trait and d = 0.31 for State, with *p* < 0.05), binge-eating (d = 1.0, *p* < 0.001), and a lower reliance on hunger/satiety signals (d = 0.99, *p* < 0.001) in comparison to those who had not been on a diet (Table 1).

## 5. Discussion

This study found that in a total sample of 405 participants, 14% were on a diet. Twice as many indicative cases with symptoms of binge-eating in the group of dieters compared to non-dieters were found. According to the EAT, when the groups were compared, the frequency of disordered eating behaviors was higher for dieters. The following results suggest a conflict between the use of food for dealing with stress and preoccupations related to body image and food that could lead to fasting. In 82% of dieters, a risk for EDs was observed (according to the EAT-26), and in these individuals, the presence of binge-eating and compensatory practices. Another highly prevalent behavior in the total sample was fasting to lose weight in 43% (*n* = 174) of participants. In this study, food consumption was not evaluated, but in Brazil, "dieting" and "weight control" have been associated with lower energy consumption and carbohydrate consumption [5].

In a recent Brazilian study, linear regression models showed "snacking" and "food liking" as being associated with increased energy intake [5]. Thus, future studies should investigate the relationship between FC resulting from negative dysphoric states during

isolation. In addition, students in different parts of the world have reported negative effects of isolation—namely, stress levels, depressive symptoms, and eating quality [5]. Bonati and colleagues have also suggested that changes in eating habits and risk for EDs may also be affected by anxiety and depression symptoms, amongst other mental health conditions, as well as by changes in sleeping patterns [23].

Different outcomes have been associated with the effects of social isolation and its adaptations to lifestyle, employment, and leisure activities [12]. In our sample, most respondents were women and were isolated from university activities, with a significant impact on food and diet practices (14% of the total number of participants). To students, fear of infection, financial uncertainty, inadequate food supplies, and absence of physical exercise had significant associations with stress, anxiety, depression, and post-traumatic symptoms [12].

Additionally, few studies have explored protective factors related to the risk of EDs and the mind–body connection. Eating patterns congruent to actual physical necessities (i.e., intuitive eating) reflect this harmonious dynamic. It could be said that resilience plays a role in the ability to contain impulses, the use of fewer emotional eating strategies, and also the ability to connect with internal signs of hunger and satiety [24]. In a representative sample in the US, individuals who ate to deal with boredom were less intuitive eaters [4]. As a result of dieting, there is less confidence in the physiological signals that regulate eating behaviors, confirmed by the lower levels of 'reliance on hunger and satiety cues' of the Intuitive Eating Scale in the present data.

Intuitive eating is a flexible eating pattern based on the promotion of a healthy connection between food and the body [25]. It primarily advocates a natural behavioral reaction to the body's states of hunger and satiety whilst deemphasizing emotional eating and behaviors; the latter two are deemed to be prejudicial to natural observations of body signs—for instance, attributing positive or negative values to foods [25]. Intuitive eating also encourages acceptance of different body shapes and sizes and regular exercising, as well as making food choices based on both their nutritional and satisfaction aspects [25,26].

Generally, diet practitioners have a diminished tendency to rely on body signs of hunger and satiety [26]. Intuitive eaters show lower levels of restrained eating, preoccupation with food (including FC), binge-eating, dieting, disordered eating, and internalization of thin standard ideals [27–29].

Finally, higher levels of cravings for food were verified in dieters (with a high effect size; Table 1). Therefore, the food microenvironment generated at the household level and the use of apps to order food are essential [5] facilitators of FC consummation, whether for food or relief of dysphoric symptoms (i.e., emotional eating). Levels of FC are influenced by the emotional state and external aspects associated with positive/negative environmental events and the presence of food alone [30].

## 6. Conclusions

This study was the first in Brazil to assess the relationship between the practice of diets in the pandemic and its impact on intuitive eating and FC. Dieting was confirmed to be associated with lower levels of confidence in internal signals, higher levels of FC, and binge-eating. Unfortunately, with the data produced in this research it was not possible to verify whether the presence of symptoms was related to the presence of a previous ED or symptoms associated with the effects of isolation.

Few studies have related FC and IE levels. Two questions from the IES-II deal with a harmonious relationship between the desire to eat and food consumption: "If I am craving a certain food, I allow myself to have it." and "I allow myself to eat what food I desire at the moment." Future studies should investigate these associations.

Considering the present data, two factors can be listed for the decrease in intuitive eating: (i) emotional factors associated with isolation and the use of food to deal with dysphoric states and (ii) the disconnection resulting from dietary patterns and rules that diets contain.

**Author Contributions:** J.O. contributed to the design of the project, analysis and interpretation of data as well as writing the article; M.B.N.S. contributed to the interpretation of data as well as writing the article; I.d.C.S. contributed to final revisions; T.A.C. contributed to the project design and final approval of the version to be published. All authors have read and agreed to the published version of the manuscript.

**Funding:** This research received no external funding.

**Institutional Review Board Statement:** The procedures were approved by the University's Institutional Review Board (2,695,532).

**Informed Consent Statement:** Informed consent was obtained from all subjects involved in the study.

**Conflicts of Interest:** The authors have no conflicts of interest to disclose relevant to this study.

**Ethical Approval:** The Psychiatric Hospital ethics committee approved the study protocol (n. 2.695.532).

## Appendix A. Supplementary Data

General participant data.

|  | *n* = 405 | % |
|---|---|---|
| **Ethnicity** |  |  |
| White | 327 | 80.7 |
| Asian | 17 | 4.2 |
| Black | 58 | 14.3 |
| Not declared | 3 | 0.7 |
| **Study area** |  |  |
| Biology | 165 | 40.7 |
| Math and Science | 81 | 20.0 |
| Human Sciences | 159 | 39.3 |
| **Body mass index** |  |  |
| Underweight | 37 | 9.1 |
| Normal weight | 239 | 59.0 |
| Overweight | 94 | 23.2 |
| Obesity I | 23 | 5.7 |
| Obesity II | 7 | 1.7 |
| Obesity III | 2 | 0.5 |
| Not related | 3 | 0.7 |
| **Eating has changed significantly** |  |  |
| No | 128 | 31.6 |
| Yes | 277 | 68.4 |
| **Are you commuting to and attending university?** |  |  |
| No | 392 | 96.8 |
| Yes | 13 | 3.2 |
| **Binge-eating frequency** |  |  |
| Twice a week or more | 42 | 10.4 |
| Less than once a week | 108 | 26.7 |
| Not once | 200 | 49.4 |
| Once a week | 55 | 13.6 |
| **Use of laxatives frequency** |  |  |
| Twice a week or more | 3 | 0.7 |
| Less than once a week | 13 | 3.2 |
| Not once | 380 | 93.8 |
| once a week | 9 | 2.2 |

|  | *n* = 405 | % |
|---|---|---|
| **Use of diuretics frequency** | | |
| Twice a week or more | 5 | 1.2 |
| Less than once a week | 4 | 1.0 |
| Not once | 390 | 96.3 |
| Once a week | 6 | 1.5 |
| **Self-induced vomiting frequency** | | |
| Twice a week or more | 6 | 1.5 |
| Less than once a week | 11 | 2.7 |
| Not once | 385 | 95.1 |
| Once a week | 3 | 0.7 |
| **Not eating or eating very little food to lose weight or not to gain weight** | | |
| Twice a week or more | 51 | 12.6 |
| Less than once a week | 85 | 21.0 |
| Not once | 231 | 57.0 |
| Once a week | 38 | 9.4 |
| **Intermittent fasting to lose weight or not to gain weight** | | |
| Twice a week or more | 42 | 10.4 |
| Less than once a week | 40 | 9.9 |
| Not once | 308 | 76.0 |
| Once a week | 15 | 3.7 |
| **Weighing before quarantine starts** | | |
| No | 128 | 31.6 |
| Yes | 277 | 68.4 |
| **Weighing during quarantine** | | |
| No | 223 | 55.1 |
| Yes | 182 | 44.9 |

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
