# Peer review of "Dieters in the Covid-19 Pandemic: Risk for Eating Disorders and Their Association with Food Cravings and Intuitive Eating Traits"

_2673-4168, doi:10.3390/obesities2030020_

Round 1

Reviewer 1 Report

The paper has been improved and is now suitable for publication

Reviewer 2 Report

Several improvements and clarifications have been made to the study.

This manuscript is a resubmission of an earlier submission. The following is a list of the peer review reports and author responses from that submission.

Round 1

Reviewer 1 Report

Comments to Authors 

            This study showed that data reinforces the high prevalence of risk for Eating Disorders in university students, the impact of dieting on FC, and intuitive eating during social isolation.

          Authors are kindly requested to emphasize the current concepts about these issues in the context of recent knowledge and the available literature. This article should be quoted in the References list.

References

  1. Changes of symptoms of eating disorders (ED) and their related psychological health issues during the COVID-19 pandemic: a systematic review and meta-analysis. J Eat Disord. 2022; 10 (1): 51. Published 2022 Apr 13. doi:10.1186/s40337-022-00550-9.
  2. Updates in the treatment of eating disorders in 2021: a year in review in Eating Disorders: The Journal of Treatment & PreventionEat Disord. 2022; 30 (2): 144-153. doi:10.1080/10640266.2022.2064109.
  3. Eating Disorders in the Time of the Covid-19 Pandemic: A Perspective [published online ahead of print, 2022 Apr 22]. Endocr Metab Immune Disord Drug Targets. 2022; doi:10.2174/1871530322666220422104009.
  4. Psychological impact of the quarantine during the COVID-19 pandemic on the general European adult population: a systematic review of the evidence. Epidemiol Psychiatr Sci. 2022;31:e27. Published 2022 Apr 27. doi:10.1017/S2045796022000051.

Reviewer 2 Report

The study has many limitations, the main one being that the actual intake of the participants was not analyzed, analyzing the diet of the individuals using food frequency consumption questionnaires.
The results are not well explained and the discussion is very poor, this should be improved.
The conclusions are not representative of the results presented. When the results are better explained, conclusions according to them can be obtained.
The idea of the work is good, but it is not well planned or developed.

  1. INSTRUMENETS: Are the questions from 1 to 8 validated? Were the answers open or closed?
  2. Criteria for selecting groups: Why were subjects who were not on a diet and had plans to lose weight discarded? Exclusion and inclusion criteria are not well explained and should be included in the Characterization of study participants section.
  3. RESULTS: The results must be explained. Don't just display a table. This needs to be modified